# Histone Acetyltransferase MOF-Mediated AURKB K215 Acetylation Drives Breast Cancer Cell Proliferation via c-MYC Stabilization

**DOI:** 10.3390/cells14141100

**Published:** 2025-07-17

**Authors:** Yujuan Miao, Na Zhang, Fuqing Li, Fei Wang, Yuyang Chen, Fuqiang Li, Xueli Cui, Qingzhi Zhao, Yong Cai, Jingji Jin

**Affiliations:** School of Life Sciences, Jilin University, Changchun 130012, China; miaoyj21@mails.jlu.edu.cn (Y.M.); zhangna@jlu.edu.cn (N.Z.); lifq22@mails.jlu.edu.cn (F.L.); fei@jlu.edu.cn (F.W.); cyy22@mails.jlu.edu.cn (Y.C.); lifq20@mails.jlu.edu.cn (F.L.) cuixl24@mails.jlu.edu.cn (X.C.); zhaoqz23@mails.jlu.edu.cn (Q.Z.)

**Keywords:** AURKB, MOF, histone acetyltransferase, acetylation, cell proliferation

## Abstract

Aurora kinase B (AURKB), a serine/threonine protein kinase, is essential for accurate chromosome segregation and cytokinesis during mitosis. Dysregulation of AURKB, often characterized by its overexpression, has been implicated in various malignancies, including breast cancer. However, the mechanisms governing its dysregulation remain incompletely understood. Here, we identify a pivotal role for the MOF/MSL complex—which includes the histone acetyltransferase MOF (KAT8)—in modulating AURKB stability through acetylation at lysine 215 (K215). This post-translational modification inhibits AURKB ubiquitination, thereby stabilizing its protein levels. MOF/MSL-mediated AURKB stabilization promotes the proper assembly of the chromosomal passenger complex (CPC), ensuring mitotic fidelity. Notably, inhibition of MOF reduces AURKB K215 acetylation, leading to decreased AURKB expression and activity. Consequently, this downregulation suppresses expression of the downstream oncogene c-MYC, ultimately attenuating the malignant proliferation of breast cancer cells. Collectively, our findings reveal a novel mechanism by which lysine acetylation regulates AURKB stability, highlight the significance of the MOF-AURKB-c-MYC axis in breast cancer progression, and suggest potential therapeutic strategies targeting this pathway in clinical settings.

## 1. Introduction

Mitosis is a fundamental biological process orchestrated by a complex network of proteins, among which Aurora kinase B (AURKB) plays a pivotal role in mitotic regulation and cell cycle progression [1]. Aurora kinases comprise a family of three highly homologous serine/threonine kinases—Aurora A, Aurora B, and Aurora C—that execute distinct yet interconnected functions in mitotic control. AURKB, also known by various aliases, including AIK2, AIM1, ARK2, AurB, IPL1, and STK5, was first identified as a cell cycle-dependent kinase in NIH3T3 cells, underscoring its role in cell proliferation [2].

AURKB is a core component of the chromosomal passenger complex (CPC) [3], functioning in concert with the scaffolding protein inner centromere protein (INCENP) and the non-enzymatic subunits Survivin (BIRC5) and Borealin (CDCA8). The CPC ensures faithful chromosome segregation by dynamically localizing to specific mitotic structures during different stages of cell division [4,5]. Precise regulation of AURKB-mediated substrate phosphorylation is essential for mitotic progression [6], particularly in early mitosis, where AURKB phosphorylates kinetochore substrates to correct improper microtubule–kinetochore attachments [7,8]. As mitosis advances, the CPC targets additional substrates to coordinate subsequent mitotic events. AURKB translocates from metaphase kinetochores to the central spindle during anaphase and later to the contractile ring during telophase [9,10,11], thereby ensuring successful cytokinesis. Aberrant AURKB expression is frequently observed in various cancers, where it is associated with chromosomal instability and increased malignancy [12]. Dysregulated AURKB is believed to confer a proliferative advantage to cancer cells [13].

MOF (males absent on the first), a histone acetyltransferase (HAT) of the MYST family, was initially identified as a component of the *Drosophila* dosage compensation complex [14,15]. In humans, MOF shares structural similarities with its *Drosophila* homolog dMOF, including a MYST catalytic domain, a chromatin-binding domain, and a C2HC-type zinc finger motif [16]. MOF forms two distinct complexes: the male-specific lethal (MSL) complex, which primarily acetylates histone H4K16, and the non-specific lethal (NSL) complex, which acetylates histones H4K5, H4K8, and H4K16 [17,18]. Beyond histone modification, MOF plays critical roles in transcriptional regulation, chromatin remodeling, cell proliferation, differentiation, and DNA damage repair [19,20,21]. Increasing evidence suggests that MOF is involved in tumorigenesis, influencing cancer cell proliferation, apoptosis, and stemness [22]. For instance, MOF overexpression in non-small cell lung cancer (NSCLC) promotes tumor progression through Nrf2 acetylation, contributing to poor prognosis and therapeutic resistance [23]. In addition, MOF-mediated acetylation of MDM2 has been implicated in cisplatin resistance in ovarian cancer [24].

Post-translational modifications (PTMs) fine-tune AURKB activity and localization throughout mitosis, with ubiquitin-mediated degradation being a well-established regulatory mechanism [25,26]. Recent studies suggest that HATs and histone deacetylases (HDACs) modulate AURKB function in cancer. In esophageal cancer, the histone acetylation reader BRD4 is recruited to the promoters of *AURKA* and *AURKB*, and its inhibition by JQ1 induces senescence [27]. In lymphoma cells, AURKB and HDACs cooperatively regulate proliferation, with inhibition of either leading to cell cycle arrest and apoptosis [28].

However, the direct acetylation of AURKB remains poorly understood. A study in 2016 reported that TIP60 (KAT5)-dependent acetylation of AURKB enhances its kinase activity, thereby ensuring proper mitotic procession and genomic stability [29]. Similarly, the MOF-containing MSL complex has been shown to modulate the stability and transcriptional activity of YY1 via acetylation [30]. Whether MOF directly acetylates AURKB, and how this modification affects AURKB stability and cancer progression, remains unknown. In this study, we demonstrate that MOF-mediated acetylation of AURKB enhances its stability and kinase activity. Specifically, MOF-driven acetylation of AURKB promotes the accumulation of c-MYC, thereby facilitating malignant proliferation in breast cancer cells.

## 2. Materials and Methods

### 2.1. Antibodies and Reagents

The following antibodies were used in this study: anti-MSL1 (mouse monoclonal, 24373-1-AP) from Proteintech (Wuhan, China); anti-MOF (rabbit polyclonal, A3390) from ABclonal Technology (Wuhan, China); anti-MSL2 (rabbit polyclonal, ab83911) from Abcam (Shanghai, China); anti-AURKB (rabbit monoclonal, AF1930) from Beyotime (Shanghai, China); anti-c-MYC (9E10), anti-INCENP (sc-376514), anti-CDCA8 (sc-376635) and anti-BIRC5 (sc-17779), anti-Akt (sc-81434), anti-mTOR (sc-517464), anti-ERK (sc-135900), anti-C-jun (sc-166540), anti-MAPK (sc-7972) (mouse monoclonal antibodies), as well as anti-CyclinB1 (sc-752), anti-E-Cadherin (sc-59778), and anti-N-Cadherin (sc-393933) (rabbit polyclonal antibodies), all from Santa Cruz Biotechnology (Dallas, TX, USA). Anti-HA (RLM3003) and anti-H3 (RLM3038) mouse monoclonal antibodies were obtained from Ruiying Biological (Suzhou, China). Anti-H3S10Ph (mouse monoclonal, #26436) was sourced from Upstate (New York, NY, USA). Anti-Flag (M2) (A2220), anti-Myc (M2)-agarose (A7470), anti-Flag M2 (mouse monoclonal, F3165), and anti-H4K16ac (H9164) (rabbit polyclonal) antibodies were obtained from Sigma (St. Louis, MO, USA). Anti-c-MYC-T58Ph (rabbit polyclonal) was from Bioss (Beijing, China). Pan-acetylation (Pan-ac, PTM0105RM) (rabbit polyclonal) antibody was from Jingjie Biotechnology (Hangzhou, China). Anti-β-Tubulin (M30109), anti-PanPh (M210030F) (mouse monoclonal), and anti-Ki67 (TW0001F) (rabbit monoclonal) antibodies were purchased from Abmart (Shanghai China). Anti-His (GB151251-100) (mouse monoclonal) antibody was from Servicebio (Wuhan, China). Anti-MSL3L1 and anti-GAPDH (rabbit polyclonal) antibodies were raised against bacterially expressed proteins at Jilin University.

The following reagents were used: cycloheximide (CHX, DH466-1) from Beijing Dingguo Changsheng Biotechnology Co., Ltd. (Beijing, China); the histone acetyltraferase (HAT) inhibitor MG149 (S7476) from Selleck Chemicals (Shanghai, China); the AURKB inhibitor AZD1152-HQPA from Abmole Bioscience (Beijing, China); and hydroxyurea (HU, H8267), Nocodazole (M1404), and MG132 (Z-Leu-Leu-al) from Sigma (St. Louis, MO, USA).

### 2.2. Cell Culture

HEK293T and HeLa cells were obtained from the Type Culture Collection of the Chinese Academy of Sciences (Shanghai, China). The breast cancer lines MCF-7 and MDA-MB-231 were purchased from the Shanghai Biotechnology Co., Ltd. (Shanghai, China). All cell lines were cultured in Dulbecco’s modified Eagle’s medium (DMEM, Meilunbio^®^, Dalian, China) supplemented with 10% fetal bovine serum (FBS, Procell, Wuhan, China) and 1% penicillin-streptomycin (P/S, Thermo Fisher Scientific, Waltham, MA, USA). Cells were maintained at 37 °C in a humidified incubator with 5% CO_2_. Each cell line was authenticated by short tandem repeat (STR) profiling within the past three years. All experiments were conducted using mycoplasma-free cells.

### 2.3. Plasmid Construction and Transfection

The coding region of full-length AURKB (NM_001313950.2), MSL1 (NM_001012241), MOF (NM_032188), INCENP (NM_020238.3), and MYC (AH002906.2), along with various truncations—including MOF (1-157aa, 1-216aa and 158-458aa)—were subcloned into pcDNA3.1(–) vectors with Flag, Myc, or HA tags. Additionally, point mutants, including AURKB (K215Q, K215R) and MOF (G327E), were generated. Plasmids were transiently transfected into cells using polyethyleneimine (PEI, 23966, PolySciences, Beijing, China) according to the manufacturer’s instructions.

### 2.4. Expression of Recombinant Proteins in Escherichia Coli

Full-length AURKB and INCENP were subcloned into pET41a vector. His-GST-tagged AURKB and INCENP proteins were expressed from pET41a vector in BL21 (DE3) Codon Plus Escherichia coli cells (Laboratory bank).

### 2.5. siRNA/shRNA Knockdown

HEK293T, MCF-7, and MDA-MB-231 breast cancer cells were transfected with non-targeting (NT) siRNA (D-001206), AURKB siRNA (#1, 5′-CCUGCGUCUCUACAACUAUtt-3′, #2, 5′-UCGUCAAGGUGGACCUAAAtt-3′), and an siRNA SMART pool (Dharmacon, Shanghai, China), using Lipofectamine RNAi MAX (13778150, Invitrogen, Shanghai, China) according to the manufacturer’s instructions. Seventy-two hours post-transfection, cells were subjected to subsequent experiments. For stable knockdown, the pLVX-shRNA system was used to express shRNA targeting AURKB, MSL1, and MOF in HEK293T, MCF-7, and MDA-MB-231 cells. The specific shRNA target sequences were as follows: shAURKB (CCUGCGUCUCUACAACUAU), shMSL1 (GCACCGGACGTGTAGGAAAT), and shMOF (CGAAATTGATGCCTGGTAT).

### 2.6. Immunoprecipitation (IP)

MCF-7, MDA-MB-231, and HEK293T cells were cultured in 10 cm tissue culture plates and transiently transfected with Flag- or Myc-tagged plasmids. Forty-eight hours post-transfection, cells were collected and lysed using RIPA buffer containing the following: 1% NP-40, 150 mM NaCl, 50 mM Tris-HCl, 10% glycerol, 1 mM dithiothreitol (DTT), and a complete protease inhibitor cocktail. Whole-cell lysates were incubated overnight at 4 °C with anti-Flag (M2) or anti-Myc-agarose beads. The immunoprecipitated proteins were then eluted using 4× SDS loading buffer and analyzed by Western blot with anti-Flag or anti-Myc antibodies.

### 2.7. Immunofluorescence Staining

HeLa and MCF-7 cells were cultured in 24-well plates containing coverslips (NEST, 8D1007, Wuxi, China) and grown to approximately 30% confluence. Cells were then transfected with plasmids and incubated for 48 h. Cells were then fixed and immunostained with primary antibodies, followed by FITC/TRITC-conjugated secondary antibodies (1:300, Santa Cruz sc-2012). Nuclei were counterstained with Vectashield containing DAPI (H-1200, Vector Laboratories, Inc., Burlingame, CA, USA). Fluorescent images were acquired using an Olympus BX40F microscope equipped with a 40× silicon immersion objective (Olympus Corporation, Miyazaki, Japan).

### 2.8. Reverse Transcription PCR

Total RNA was extracted using RNAiso Plus (9109; Takara, Tokyo, Japan). An amount of 1 μg of total RNA from each sample was reverse transcribed into cDNA using the PrimeScript 1st Strand synthesis Kit (6110A, Takara, Tokyo, Japan). Relative mRNA levels were quantified using TB Green^®^ Premix Ex Taq ™ II (RR820A, Takara, Tokyo, Japan) on the Eco Real-Time PCR System (Illumina, Gene Company Limited, Hong Kong, China). The qPCR primers were as follows: AURKB, 5′-CAGTGGGACACCCGACATC -3′ (forward) and 5′-GTACACGTTTCCAAACTTGCC -3′ (reverse); MSL1, 5′- CAAGACTCTCCACTCCCCAAAA -3′ (forward) and 5′- CCTCCAAGAAGGAATTGCTACAG -3′ (reverse); MOF, 5′- CCCAAACCAGTCAGACCAGC -3′ (forward) and 5′- GGGCCACCAGAACTGACTTT -3′ (reverse); GAPDH, 5′-ATCACTGCCACCCAGAAGAC-3′ (forward) and 5′-ATGAGGTCCACCACCCTGTT-3′ (reverse).

### 2.9. In Vitro KAT Assay

The process was carried out as follows. Mix the following reactants to reach a total volume of 20 μL: cold acetyl coenzyme A (12.5 μM), recombinant AURKB proteins expressed in Escherichia coli, anti-Flag MOF beads, HAT reaction buffer. The HAT reaction buffer contains 50 mM Tris-HCl (pH 8.0), 50 mM KCl, 0.1 mM EDTA, 1 mM dithiothreitol, 5% (*v*/*v*) glycerol, 1 mM phenylmethylsulfonyl fluoride, 10 mM sodium butyrate. Incubate the reaction mixture at 30 °C for 30 to 60 min. Finally, add 4× SDS loading buffer to terminate the reaction, and heat the sample at 95 °C for Western blot analysis.

### 2.10. Flow Cytometry Analysis

MCF-7 and MDA-MB-231 cells were cultured in DMEM supplemented with 10% FBS. For fixation, cells were harvested and resuspended as single cells in 70% ethanol at 20 °C for at least 4 h. Following ethanol fixation, cells were centrifuged at 300× *g* for 5 min, and the supernatant was discarded. The cell pellets were washed and resuspended in 300 μL PBS containing 0.1% (*v*/*v*) Triton X-100 (Sigma, Cat. T8787), 0.3 mg/mL DNase-free RNase A (Sigma, Cat. R5500), and 50 μg/mL propidium iodide (CF0031, Beijing Dingguo, China), followed by incubation at 37 °C for 1 h. Flow cytometry was performed using an EPICS XL ™ cytometers (Beckman Coulter, Life Sciences, Shanghai, China), and data were analyzed with ModFit LT 5.0 software (Verity Software House, MA, USA).

### 2.11. EdU Assay

The EdU incorporation assay was performed using the BeyoCleck™ EdU Cell Proliferation Kit (Beyotime, C0071s, Shanghai, China) and the Alexa Fluor 488 in vitro Imaging Kit (Abcam, ab150165, Shanghai, China). MCF-7 and MDA-MB-231 cells were incubated with 10 µM EdU (5-ethyl-2′-deoxyuridine) at 37 °C for 2 h. Cells were then fixed with 4% paraformaldehyde for 15 min and washed with PBS containing 0.5% Triton-X-100. Nuclei were counterstained with Hoechst 33342 (GC10939, GLPBIO, Shanghai, China). The proliferation rate was determined according to the manufacturer’s instructions. Fluorescent images were acquired using a fluorescence microscope, with three randomly selected fields captured per group.

### 2.12. Cell Viability Assay

MCF-7 and MDA-MB-231 cells (1000 cells/well) were seeded in 96-well plates, and cell viability was assessed using the CellTiter 96^®^ Aqueous One Solution Cell Proliferation Assay Kit (G3580, Promega Corporation, Madison, WI, USA) at 24, 48, 72, and 96 h. Absorbance was measured at 490 nm using a microplate reader (Infinite F200 Pro, TECAN, Shanghai, China).

### 2.13. Colony Formation Assay

Stably transferred MCF-7 and MDA-MB-231 cell lines (3000–4000 cells/well) were seeded into 6-well plates and cultured for 10–14 days at 37 °C. Colonies were then fixed with 4% paraformaldehyde for 15 min and stained with 0.1% crystal violet for 20 min. The number and size of colonies were recorded for comparison and imaged using a digital camera.

### 2.14. In Vivo Tumor Metastasis Experiments

All animal studies were approved by the Institutional Animal Care and Use Committee (IACUC) of Jilin University. Male Balb/c mice (7–8 weeks old, weighing 18–20 g) were purchased from Vital River Biotechnology Company (Beijing, China). The animal experiment protocol was approved under the ethics review number (2024) YNPZSY No. (0512). Mice were housed under a 12 h light/dark cycle with ad libitum access to food and water. Randomization was performed based on body weight, and sample sizes were determined according to the “Resource Equation” method. MDA-MB-231 cells infected with lentiviral-mediated pLVX-Flag-AURKB-WT (*n* = 5) and pLVX-Flag-AURKB K215R (*n* = 5) were subcutaneously injected into mice. Thirty days post-injection, mice were euthanized, and tumor tissues were collected for further analysis.

### 2.15. Statistical Analysis

Statistical analyses were conducted using data from at least three independent experiments. Data were processed with SPSS software, version 26 (IBM Corp., Armonk, NY, USA). Results are presented as mean ± SD. Differences between two groups were evaluated using an unpaired Student’s *t*-test, while comparisons among multiple groups were analyzed using one-way analysis of variance (ANOVA). A *p*-value < 0.05 was considered statistically significant.

## 3. Results

### 3.1. A Reciprocal Interaction Between the MOF/MSL Complex and CPC in HEK293T Cells

Previously, we generated an MSL1-knockout (MSL1-KO) HEK293T cell line using CRISPR/Cas9 gene editing [31]. To investigate the regulatory function of the MSL complex, we performed SILAC-based mass spectrometry on MSL1-KO cells. Among 4449 differentially expressed proteins (Figure 1A), several chromosomal passenger complex (CPC) subunits—AURKB, INCENP, and BIRC5—were significantly downregulated (fold change > 1.2, *p* < 0.05) (Figure 1B), suggesting a regulatory link between the MOF/MSL complex and the CPC. To validate this hypothesis, we overexpressed Flag-tagged MOF or MSL1 in HEK293T cells and observed a dose-dependent increase in endogenous AURKB protein levels, accompanied by elevated phosphorylation of its substrate histone H3 at serine 10 (H3S10) (Figure 1C). Conversely, shRNA-mediated knockdown of MOF significantly reduced AURKB expression and H3S10 phosphorylation (Figure 1D). Notably, quantitative PCR (qPCR) analysis confirmed that neither overexpression nor knockdown of MOF nor MSL1 affected AURKB mRNA levels (Figure 1E,F), suggesting that the MOF/MSL complex regulates CPC components at the post-transcriptional level. Consistent with this, MSL1-KO cells exhibited reduced expression of all CPC subunits, along with decreased H3S10 phosphorylation (Figure 1G, lane 2 vs. lane 1). Importantly, reintroduction of Flag-MSL1 into MSL1-KO cells restored the expression of CPC components—including AURKB, BIRC5, CDCA8, and INCENP—along with H3S10 phosphorylation (Figure 1G, lane 4 vs. lane 3). The quantitative results are shown below the corresponding protein bands.

To further examine the role of the MOF/MSL1 complex in regulating AURKB protein stability, we conducted a cycloheximide (CHX) chase assays. AURKB degradation was markedly accelerated in HEK293T cells treated with shMOF (Figure 1H,I, lanes 7–12) or shMSL1 (Figure 1J,K, lanes 7–12) compared to cells expressing non-targeting shRNA (shNT), confirming that MOF/MSL1 depletion shortens the half-life of AURKB. Finally, an in vitro phosphorylation assay demonstrated that AURKB catalytic activity toward its substrate INCENP (Figure 1L, lane 4) was significantly enhanced in the presence of MSL1 (Figure 1M, lane 3).

Collectively, these findings establish a reciprocal relationship between the MOF/MSL complex and the CPC, wherein the MOF/MSL complex stabilizes AURKB protein and enhances CPC activity, thereby supporting proper mitotic regulation.

### 3.2. MOF, MSL1, and AURKB May Collaborate During Early Mitosis

Given the regulatory effects of the MOF/MSL complex on AURKB kinase activity and CPC subunit expression, we hypothesized that MOF and MSL1 physically interact with AURKB. Flag IP assays confirmed this interaction (Figure 2A–C), showing that endogenous MOF, MSL1, and AURKB co-precipitated with Flag-tagged AURKB, MOF, and MSL1, respectively, indicating a direct binding affinity among these proteins. This interaction was further validated by co-transfection of MOF and AURKB in HEK293T cells, followed by co-IP experiments, which confirmed their association (Figure 2D,E, lane 3).

To delineate the minimal binding region of MOF required for interaction with AURKB, we co-transfected Flag-AURKB with the full-length Myc-tagged MOF (1-458 aa) and a series of truncated MOF mutants: Myc-MOF-F1 (1-157 aa), HA-MOF-L1 (1-216 aa), and HA-MOF-L2 (158-458 aa) (Figure 2F, upper panel). Flag IP followed by Western blot analysis revealed that the C-terminal region of MOF (216-458 aa) was essential for AURKB binding (Figure 2G, lower panel), suggesting that the histone acetyltransferase (HAT) domain of MOF mediates this interaction.

Given the dynamic localization of CPC components throughout the cell cycle—and the phase-specific catalytic activity of AURKB [5]—we next investigated the subcellular co-localization of MOF, MSL1, and AURKB during mitosis. Immunofluorescence (IF) staining was performed in HeLa cells using β-tubulin (red) as a marker to distinguish prophase, metaphase, anaphase, and telophase. AURKB, MOF, and MSL1 (green) co-localized on chromosomes during prophase and metaphase. However, during anaphase, while MOF and MSL1 remained chromosomally associated, AURKB translocated to the equatorial plate and subsequently to the midbody during telophase, consistent with its role in cytokinesis Figure 2H–J). These findings suggest that the MOF/MSL complex is involved in regulating AURKB function during early mitosis.

To further validate the interaction between the MOF/MSL and CPC complexes, we co-transfected HEK293T cells with Flag-tagged INCENP, a scaffolding component of the CPC, along with MOF or MSL1. Co-IP assays confirmed that both MOF and MSL1 interacted with Flag-INCENP, with endogenous AURKB detected in the immunoprecipitates (Figure 2K, lane 4 vs. lane 2; Figure 2L, lane 3 vs. lane 1). In contrast, MSL1-knockdown (KD) markedly reduced the co-IP of CPC subunits with Flag-INCENP (Figure 2M, lane 4 vs. lane 2), further supporting the role of the MOF/MSL complex in CPC stabilization. Quantified protein levels are shown below the corresponding protein bands.

### 3.3. MOF/MSL Complex Mediates AURKB Acetylation, Stabilizing CPC Integrity in HEK293T Cells

The observed interaction between MOF, MSL1, and AURKB prompted us to investigate whether the MOF/MSL complex directly acetylates AURKB and how this modification affects its protein stability. To address this, Flag-AURKB was co-transfected with Myc-tagged MOF or MSL1 into HEK293T cells, followed by Flag IP in the presence of HA-ubiquitin and the proteasome inhibitor MG132. As expected, both MOF (Figure 3A) and MSL1 (Figure 3B) bound to and acetylated AURKB (Flag IP/Pan-ac). Notably, acetylation of AURKB by the MOF/MSL complex inhibited its ubiquitination (Flag IP/HA), suggesting that this modification stabilizes AURKB by preventing proteasome-mediated degradation. Consistently, MOF knockdown reduced AURKB acetylation while increasing its ubiquitination, further supporting the role of MOF in AURKB stabilization (Figure 3C, Flag IP/Pan-ac and HA). In addition, co-transfection with an enzymatically inactive MOF mutant (G327E), which lacks acetyltransferase activity [32], significantly reduced AURKB binding and acetylation (Figure 3D, lane 6 vs. lane 4). These findings were corroborated by an in vitro lysine acetyltransferase (KAT) assay (Figure 3E, upper panel), in which purified His-tagged AURKB from *E. coli* served as the substrate (Figure 3E, lane 5).

To further dissect the functional consequences of MOF/MSL complex-mediated AURKB acetylation on protein stability and kinase activity, a series of cellular assays were performed. As shown in Figure 3F,G, co-expression of Flag-AURKB and MOF in the presence of either MSL1 overexpression or knockdown revealed that MSL1 enhances AURKB acetylation (Flag IP/Pan-ac), reduces its ubiquitination (Flag IP/HA), and promotes H3S10 phosphorylation (Input/H3S10p). Conversely, MSL1 knockdown abrogated these effects, suggesting that MSL1 acts as a cofactor to support MOF-dependent acetylation and stabilization of AURKB. Furthermore, co-expression of Myc-AURKB and Flag-INCENP in MSL1-KO HEK293T cells showed that MSL1 deletion impaired AURKB-INCENP interaction, reduced AURKB acetylation, and accelerated its degradation (Figure 3H, lane 2 vs. lane 1). These results indicate that MSL1 is required for efficient MOF-mediated regulation of CPC stability through AURKB. Collectively, our results suggest that MOF/MSL complex-mediated acetylation of AURKB is critical for maintaining CPC stability.

### 3.4. MOF/MSL Complex-Mediated Acetylation of AURKB at K215 Maintains CPC Integrity and Kinase Activity

Our findings demonstrate that MOF/MSL complex-mediated acetylation positively regulates AURKB kinase activity and stabilizes the CPC complex. To identify the specific acetylation site on AURKB, we analyzed its predicted three-dimensional structure in complex with MOF using data from the RCSB Protein Data Bank (https://www.rcsb.org/, accessed on 13 October 2023) and performed protein–protein docking via the HDOCK Server. This analysis identified lysine 215 (K215) of AURKB as a key interaction site, forming a hydrogen bond with aspartic acid 254 (D254) of MOF, with a bond length of 3.2Å and a docking confidence score of 0.9239, suggesting a stable interaction (Figure 4A). Consistent with this prediction, high-throughput mass spectrometry data from PhosphoSitePlus (https://www.phosphosite.org, accessed on 18 November 2020) also identified K215 as a putative acetylation site (Figure 4B).

To experimentally validate K215 as the primary MOF-targeted acetylation site on AURKB, we performed Flag IP in HEK293T cells co-transfected with Myc-MOF and either wild-type or K215R mutant Flag-AURKB. Western blot analysis revealed that mutation of K215 abolished MOF-mediated acetylation of AURKB (Figure 4C, Flag IP/Pan-ac), confirming K215 as the principal acetylation site. A CHX chase assay further demonstrated that the half-life of AURKB was markedly reduced upon K215 mutation (Figure 4D, lanes 7–12 vs. lanes 1–6, quantified in lower panel). Moreover, co-transfection with HA-ubiquitin and MG132 showed that the K215R mutant underwent significantly enhanced ubiquitin-mediated degradation compared to wild-type AURKB and acetylation-mimetic K215Q AURKB (Figure 4E, lane 4 vs. lanes 2 and 3).

To further assess the role of MOF-mediated acetylation in AURKB stabilization, Flag-tagged wild-type or K215R AURKB was transfected into HEK293T cells, with or without Myc-MOF co-expression. Flag IP results showed that K215 mutation accelerated AURKB degradation through the proteasome pathway (Figure 4F, Flag IP/HA, lanes 3 and 5). Additionally, the K215R mutant exhibited reduced INCENP phosphorylation (Figure 4G, lane 4 vs. lane 3), whereas the acetylation-mimetic K215Q mutant led to increased levels of phospho-INCENP (Lane 5 vs. Lane 3). Beyond CPC activity, the K215 mutation impaired AURKB’s ability to maintain CPC integrity, as evidenced by decreased expression of CPC components INCENP, CDCA8 and BIRC5 (Figure 4H, Flag IP/lane 3 vs. lanes 2 and 4), along with attenuated AURKB kinase activity, reflected by lower levels of H3S10 phosphorylation (Input/H3S10p). Notably, the K215 mutation also reduced total c-MYC protein levels while increased c-MYC phosphorylation at T58 (Input/c-MYC, Input/c-MYC-T58p), a modification known to promote c-MYC degradation. In summary, MOF-mediated acetylation of AURKB at K215 (Figure 4I) is crucial for stabilizing the CPC complex, sustaining AURKB kinase activity, and regulating downstream targets as c-MYC.

### 3.5. MOF/MSL1 Complex-Mediated Acetylation of AURKB at K215 Regulates G2/M Phase Progression in HeLa and MCF-7 Cells

Our findings indicate that the MOF/MSL complex plays a crucial role in mitotic progression by acetylating AURKB at K215. This post-translational modification is essential for maintaining the integrity and stability of the CPC, thereby ensuring proper spindle formation and chromosome segregation. To investigate the role of MOF and MSL1 in mitosis, we performed IF staining in HeLa cells following knockdown of MOF or MSL1. Compared to cells transfected with non-targeting shRNA (shNT), cells treated with shMOF or shMSL1 exhibited a significant increase in spindle multipolarity, a hallmark of mitotic disruption (Figure 5A,B). Notably, this phenotype was rescued by ectopic expression of AURKB (Figure 5C, shMOF + AURKB vs. shMOF). A comparable phenotype was observed in cells expressing the acetylation-deficient AURKB K215R mutant (Figure 5D), suggesting that MOF/MSL1 regulates mitotic progression through AURKB K215 acetylation. Quantification of spindle multipolarity (Figure 5E–H) further corroborated these findings. Western blot analysis confirmed efficient knockdown of MOF and MSL1 (Figure 5I, lanes 2 and 4; Figure 5J, lane 2–4), correlating their depletion with mitotic abnormalities.

To further assess the impact of AURKB K215 acetylation on cell cycle progression, HeLa cells were transiently transfected with either wild-type or K215R-mutant AURKB, synchronized at G1/S using 1 mM hydroxyurea, and subsequently released. Flow cytometry analysis confirmed that cells expressing the K215R mutant progressed more slowly through the G2/M phase transition compared to those expressing wild-type AURKB (Figure 5K). Western blot analysis of samples collected at 0, 6, 9, and 12 h post-release showed distinct expression kinetics: wild-type AURKB peaked at 9 h, whereas the K215R mutant exhibited a delayed peak at 12 h (Figure 5L), further underscoring the importance of K215 acetylation in timely mitotic entry.

In both MCF-7 and MDA-MB-231 breast cancer cells, AURKB depletion led to G2/M phase arrest. Reintroduction of wild-type (WT) or K215Q AURKB restored normal cell cycle progression, whereas the K215R mutant failed to do so (Figure 5M). Phase-specific analysis of the cell cycle (G1, S, G2/M, and M phases) revealed that MOF, MSL1, and H4K16ac levels were highest during G2/M and M phases (Figure 5N, lane 3), aligning with AURKB expression and reinforcing their role in CPC regulation. Moreover, analysis of The Cancer Genome Atlas (TCGA) database (https://ualcan.path.uab.edu, accessed on 10 July 2025) revealed a positive correlation between AURKB and MSL1/MSL3 expression in breast cancer (Figure 5O).

Collectively, these findings provide compelling evidence that the MOF/MSL complex acetylates AURKB at K215, stabilizing the CPC, enhancing AURKB kinase activity, and ensuring proper spindle assembly and chromosomal alignment during early mitosis. The delayed mitotic progression observed in K215R-expressing cells highlights the critical role of this acetylation event in maintaining mitotic fidelity.

### 3.6. MOF-Mediated Acetylation of AURKB at K215 Is Essential for Breast Cancer Cell Proliferation

Given that MOF-mediated acetylation of AURKB at K215 is implicated in early mitosis, we hypothesized that MOF may regulate cancer cell proliferation through AURKB. As expected, EdU incorporation assays revealed that MOF depletion significantly reduced DNA replication, indicating impaired cell proliferation. Notably, cells expressing wild-type (WT) or acetylation-mimetic AURKB (K215Q) exhibited near-normal EdU incorporation, whereas cells expressing the K215R mutant displayed markedly reduced incorporation (Figure 6A,B). Consistent with these findings, CCK-8 cell viability assays demonstrated a significant decrease in cell viability following MOF depletion, which was rescued by overexpression of WT or K215Q AURKB, but not by the K215R mutant (Figure 6C). Similarly, clonogenic assays revealed a substantial reduction in colony-forming ability upon MOF knockdown in both MCF-7 and MDA-MB-231 cells (Figure 6D,E). This defect was rescued by WT or K215Q AURKB, but not by the K215R mutant, further supporting the functional relevance of AURKB K215 acetylation in proliferation.

To further examine the role of AURKB K215 acetylation in tumorigenesis, we established stable MDA-MB-231 cell lines expressing either Flag-AURKB or Flag-AURKB K215R mutants (Figure 6F). To assess the in vivo relevance of MOF-mediated AURKB acetylation, we employed a xenograft model in which BALB/c nude mice were subcutaneously injected with these stable cell lines (*n* = 5 per group) (Figure 6G). While no significant difference in body weight was observed over the 30-day period, tumor growth was markedly reduced in the K215R group compared to the WT group (Figure 6H), suggesting that AURKB K215 acetylation promotes MDA-MB-231 proliferation by mediating. At the end of the study, tumors were excised, photographed (Figure 6I), and weighted. Tumor from the K215R group were significantly smaller (Figure 6J), confirming a diminished tumorigenic capacity in the absent of AURKB acetylation. Western blot analysis of whole-cell lysates revealed reduced levels of Ki67 and H3S10p in the K215R group (Figure 6K), consistent with impaired proliferation. Interestingly, we also observed reduced c-MYC expression and increased phosphorylation at c-MYC-T58—a modification associated with proteasomal degradation—in the K215R group, suggesting a potential link between the MOF-AURKB axis and c-MYC regulation.

Together, these finding demonstrated that MOF regulates breast cancer cell proliferation and tumor growth via AURKB K215 acetylation and raises the possibility of a broader oncogenic role for this epigenetic axis, warranting further investigation.

### 3.7. Acetylation of AURKB at K215 Promotes Breast Cancer Cell Proliferation by Stabilizing c-MYC

Xenograft studies suggest that AURKB plays a regulatory role in c-MYC expression and phosphorylation. To elucidate the molecular mechanism by which MOF-mediated acetylation of AURKB at K215 promotes breast cancer cell proliferation, we first examined the expression levels of key oncogenic proteins, particularly c-MYC, following AURKB knockdown in MCF-7 and MDA-MB-231 cells. Notably, depletion of AURKB resulted in a marked reduction in c-JUN, ERK, c-MYC, and H3S10p levels in both cell lines (Figure 7A,B, lane 2). To further delineate the AURKB regulatory network, we performed an integrated Venn analysis incorporating multiple datasets: CRISPR screen data from the BioGRID ORCS database (https://orcs.thebiogrid.org, accessed on 9 December 2023) for MCF-7 and MDA-MB-231 cell proliferation, protein interaction data from the STRING database (https://cn.string-db.org/, accessed on 9 December 2023), and genes positively associated with AURKB in breast cancer from the TCGA database (https://ualcan.path.uab.edu, accessed on 9 December 2023). This analysis identified four overlapping proteins—c-MYC, BIRC5, CDC20 and NUF2 (Figure 7C,D)—suggesting that AURKB positively regulates these proliferation-associated factors.

c-MYC is a well-established oncogene transcription factor that governs tumor cell proliferation, apoptosis, and metabolic reprogramming [33]. AURKB-mediated phosphorylation of c-MYC has been reported to inhibit its ubiquitination, thereby stabilizing the protein and promoting tumorigenesis [34]. In line with this, inhibition of AURKB with the AURKB kinase inhibitor AZD1152 at concentrations of 10, 20, and 40 nM in MCF-7 cells led to a significant reduction in Ki67 and H3S10p levels, accompanied by a dose-dependent decrease in c-MYC protein levels (Figure 7E). To determine whether MOF is implicated in AURKB-mediated c-MYC accumulation, we treated MCF-7 cells with the MOF enzymatic activity inhibitor MG149 (20, 30, and 40 μM) for 24 h. Western blot analysis revealed a dose-dependent decrease in AURKB and c-MYC expression, as well as reduced Ki67 levels, suggesting that MOF activity is required for maintaining c-MYC levels and cells proliferation (Figure 7F). To further confirm the specific role of MOF-mediated AURKB K215 acetylation in regulating c-MYC, we performed rescue experiments in AURKB-depleted MCF-7 and MDA-MB-231 cells by expressing WT AURKB, the acetylation-deficient K215R mutant, and the acetylation-mimetic K215Q mutant. As expected, AURKB knockdown significantly decreased c-MYC protein levels (Figure 7G,I, lanes 2–3), which is quantified in Figure 7H,J. This effect was rescued by expression of WT or K215Q AURKB, but not by K215R, indicating that K215 acetylation is essential for c-MYC stabilization. To validate the role of the MOF-AURKB axis in c-MYC regulation, we overexpressed WT, K215R, and K215Q AURKB in MOF-knockdown MCF-7 and MDA-MB-231 cells. Only WT and K215Q AURKB restored c-MYC expression, while the K215R mutant failed to do so (Figure 7K–N), further demonstrating that acetylation at K215 is critical for AURKB’s ability to regulate c-MYC. In summary, these findings demonstrate that acetylation of AURKB at K215 stabilizes c-MYC and may present a key mechanism through which MOF-mediated AURKB acetylation promotes breast cancer cell proliferation.

## 4. Discussion

RNAi screening has identified MOF as a critical regulator of cancer cell survival. MOF has been previously implicated in the regulation of the G2/M cell cycle checkpoint, underscoring its essential role in tumor biology and highlighting its potential as a therapeutic target [35]. Another key oncogene, AURKB, is widely recognized as a therapeutic target across multiple cancer types. AURKB inhibitors have shown enhanced efficacy when combined with osimertinib in the treatment of non-small cell lung cancer (NSCLC) [36]. Our study provides novel mechanistic insights into the molecular interplay between the MOF/MSL complex and the CPC complex. We demonstrate, for the first time, that MOF directly acetylates AURKB, thereby reducing its ubiquitination and enhancing its kinase activity. Furthermore, the MSL1 subunit reinforces the interaction between MOF and AURKB, stabilizing the complex during the G2/M phase. Importantly, acetylation of AURKB at K215 is indispensable for mitotic progression. In breast cancer cells, MOF-mediated acetylation of AURKB enhances AURKB protein stability and promotes c-MYC phosphorylation, ultimately leading to c-MYC accumulation and increased tumor cell proliferation.

The CPC plays a crucial role in regulating mitosis, with AURKB serving as its catalytic core. AURKB is essential for proper kinetochore assembly, spindle microtubule alignment, and correction of aberrant microtubule–kinetochore attachments [37,38]. Proteomic profiling of MSL1-deficient cells revealed differential expression of NUF2, a subunit of the NDC80 (nuclear division cycle 80) complex, and a well-established AURKB substrate [39]. AURKB modulates the microtubule-binding affinity of the NDC80 complex through site-specific phosphorylating NDC80, modulating kinetochore–microtubule interactions [40]. Given that AURKB activity is tightly regulated by post-translational modifications, our data suggest that MOF-mediated acetylation serves as an additional regulatory layer that modulates both AURKB stability and kinase activity.

Notably, we observed co-localization of the MOF/MSL complex and the CPC at the equatorial plate during mid-mitosis, suggesting that MOF/MSL-mediated acetylation may dynamically regulate AURKB function during chromosome segregation. Further high-resolution structural studies, such as cryo-electron microscopy [41], will be necessary to precisely define the molecular interface between MOF/MSL and AURKB during mitosis. Our findings support a model in which MSL1 is a scaffold protein that stabilizes the MOF/MSL complex and enhances its acetyltransferase activity, thereby facilitating AURKB acetylation. Loss of MSL1 results in reduced AURKB acetylation and consequently decreases H3S10 phosphorylation by AURKB. The CPC complex comprises AURKB, INCENP, BIRC5, and CDCA8. Acetylated AURKB promotes the phosphorylation of INCENP, thereby amplifying CPC kinase activity.

AURKB has garnered significant attention in cancer research due to its elevated expression in multiple malignancies, including colorectal adenocarcinoma, thyroid follicular carcinoma, laryngeal carcinoma, and lung cancer [42]. In breast cancer, AURKB expression is markedly upregulated and correlates with increased tumor cell proliferation and resistance to therapy [43]. Furthermore, AURKB overexpression has been implicated in paclitaxel resistance in NSCLC [44] and is associated with poor prognosis in hematological malignancies, such as acute lymphoblastic leukemia and acute myeloid leukemia [45]. Similarly, in hepatocellular carcinoma, AURKB mRNA levels are significantly elevated in tumor tissues and serve as an independent prognostic marker for disease aggressiveness [46]. Importantly, our study reveals that the MOF/MSL complex promotes breast cancer cell proliferation by stabilizing the CPC through acetylation of AURKB at K215—without altering AURKB mRNA expression—highlighting the significance of post-translational regulation in tumor progression.

The c-MYC oncogenes family (MYC, MYCN, MYCL) plays a pivotal role in tumorigenesis, particularly in cancers characterized by MYC overexpression or amplification [47]. Dysregulated MYC activity is a hallmark of various malignancies, including breast cancer, where it is linked to aggressive disease progression [48]. Clinical evidence indicates that c-MYC is frequently overexpressed in breast tumors and contributes to both oncogenic transformation and tumor maintenance [49,50]. Although AURKB inhibition has shown therapeutic potential in combination regimens for breast cancer, the emergence of resistance remains a significant clinical challenge [51]. Thus, dual targeting of MOF and AURKB may provide a promising strategy to suppress tumor proliferation, potentially reducing chemotherapy dosage requirements and mitigating resistance. In this context, our findings underscore the importance of combined inhibition strategies involving AURKB and highlight its critical role in drug-resistant breast cancer.

## 5. Conclusions

Our study demonstrates that MOF-mediated acetylation of AURKB at K215 enhances AURKB stability and kinase activity, thereby promoting c-MYC accumulation and supporting breast cancer cell proliferation. Given the central role of this regulatory axis in tumor progression, targeting MOF and AURKB represents a compelling therapeutic strategy. Pharmacological inhibition of MOF or AURKB may effectively disrupt this oncogenic circuit, suppress tumor growth, and offer a novel avenue for intervention. Future studies should focus on the development of specific inhibitors that target MOF-AURKB interactions, as well as evaluate their therapeutic potential in combination with existing anti-cancer regimens.

## Figures and Tables

**Figure 1 cells-14-01100-f001:**
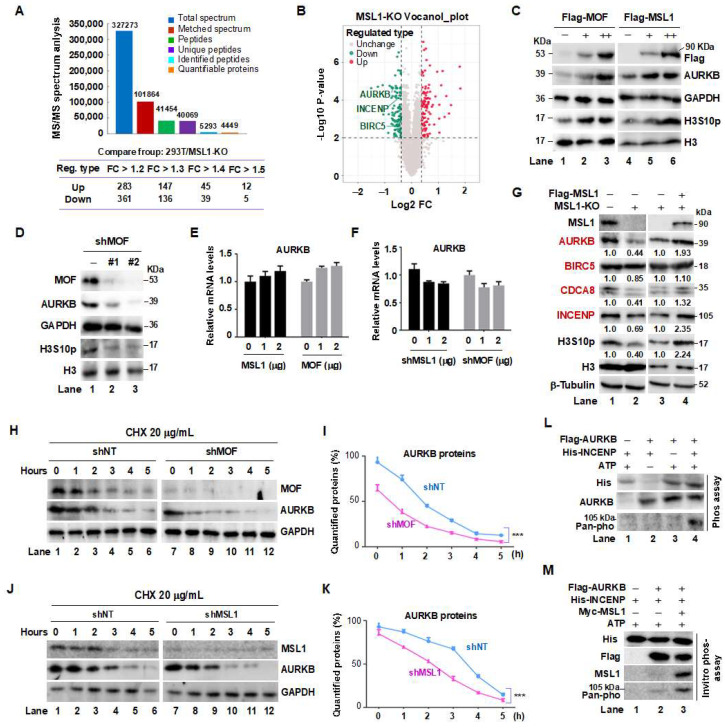
Crosstalk between the MOF/MSL complex and CPC in HEK293T cells. (**A**) Summary of SILAC- based mass spectrometry data from MSL1-knockout (MSL1-KO) cells. (**B**) Volcano plot of protein expression changes in MSL1-KO cells, highlighting the significant downregulation of CPC subunits AURKB, INCENP, and BIRC5. (**C**) Overexpression of MOF or MSL1 increases endogenous AURKB protein levels and kinase activity. HEK293T cells were transfected with Flag-tagged MOF or MSL1 for 48 h, and AURKB levels along with H3S10 phosphorylation (H3S10p) were assessed by Western blot. (**D**) Knockdown of MOF reduces AURKB protein levels and H3S10p. (**E**,**F**) Relative AURKB mRNA levels, measured by RT-qPCR, remained unchanged following MOF or MSL1 overexpression or knockdown. GAPDH was used for normalization. (**G**) Overexpression of MSL1 in MSL1-KO cells restores protein levels of AURKB, BIRC5, CDCA8, and INCENP (red). (**H**–**K**) Assessment of AURKB protein half-life. HEK293T cells transfected with shNT, shMOF, or shMSL1 were treated with 20 μg/mL CHX, and AURKB protein levels were analyzed at 0, 1, 2, 3, 4 and 5 h. (**L**) In vitro phosphorylation assay: Flag immunoprecipitation (IP) was performed in cells overexpressing Flag-AURKB, followed by an in vitro kinase assay using Flag-IP eluates, *E. coli*-expressed/purified His-INCENP, and ATP. Phosphorylated INCENP was detected using an anti-pan-phospho antibody. (**M**) Overexpression of Myc-tagged MSL1 enhances INCENP phosphorylation. GAPDH, β-tubulin, and H3 were used as internal loading controls throughout the figure. *** *p* < 0.001 compared to shNT group, Student’s *t*-test.

**Figure 2 cells-14-01100-f002:**
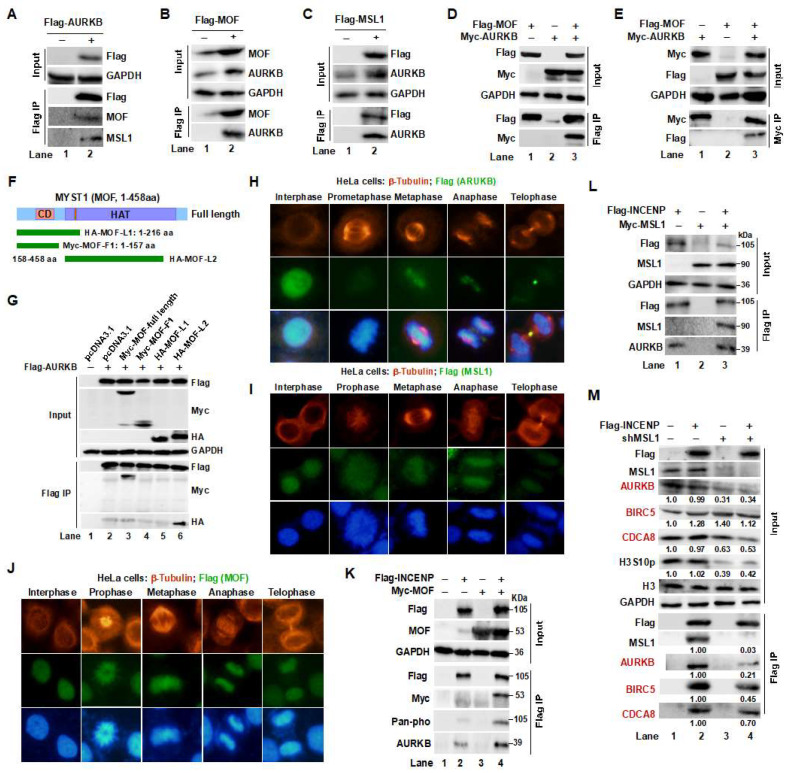
Interaction and co-distribution of MOF/MSL1 and AURKB during mitosis in HeLa cells. (**A**–**C**) Interaction between the MOF/MSL complex and AURKB. Flag IP was performed in HEK293T cells expressing Flag-tagged AURKB, MOF, or MSL1. Endogenous MOF, MSL1, and AURKB proteins were detected in the IP eluates, confirming their physical association. (**D**,**E**) Co-transfection and Co-IP further validate the interaction between AURKB and the MOF/MSL complex in HEK293T cells. (**F**,**G**) Domain mapping of MOF required for AURKB binding. MOF truncation mutants were generated based on distinct structural domains (upper panel). Full-length Flag-AURKB and MOF deletion constructs were co-transfected into 293T cells. After 48 h, Flag IP followed by Western blot analysis revealed that the C-terminal region of MOF is necessary for AURKB interaction (lower panel). (**H**–**J**) IF analysis of AURKB, MOF, and MSL1 localization during different mitotic phases. HeLa cells were stained with antibodies against AURKB, MOF, and MSL1 (green), β-tubulin (red) and DAPI (blue) to indicate nuclei. Scale bar: 20 μm. Images were captured with a 40× objective. (**K**,**L**) The MOF/MSL1 complex interacts with the CPC. HEK293T cells were co-transfected with Flag-INCENP and Myc-tagged MOF or MSL1. Flag IP was performed 48 h post-transfection and bound proteins—including phosphorylated-INCENP—were detected by Western blot. (**M**) MSL1 knockdown destabilizes the CPC complex. Flag IP was conducted in cells expressing Flag-INCENP with or without shMSL1. Input and IP fractions were analyzed by Western blot to assess levels of CPC components (red).

**Figure 3 cells-14-01100-f003:**
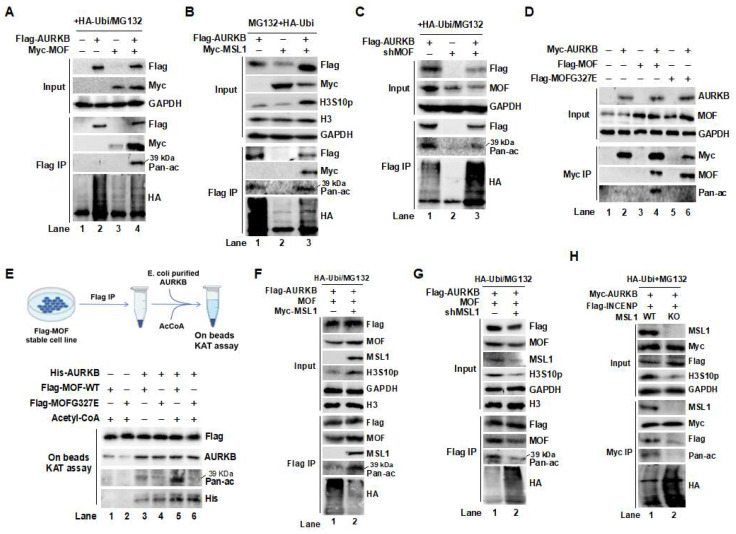
Acetylation of AURKB by the MOF/MSL1 complex stabilizes the CPC in HEK293T cells. (**A**,**B**) The MOF/MSL1 complex promotes AURKB acetylation and stability. HEK293Tcells were co-transfected with Flag-AURKB with Myc-tagged MOF or MSL1 in the presence of HA-Ubiquitin and the proteasome inhibitor MG132. Following Flag IP, acetylation and ubiquitination of AURKB were analyzed by Western blot. (**C**) Knockdown of MOF (shMOF) reduces AURKB acetylation and increases its ubiquitination, indicating decreased protein stability. (**D**) The enzymatically inactive MOF G327E mutant fails to acetylate AURKB, as shown by Flag IP followed by Western blot. (**E**) In vitro lysine acetyltransferase (KAT) assay. Recombinant His-AURKB protein was expressed and purified from *E. coli*. The KAT assay was performed by incubating His-AURKB with Flag IP eluates from MOF-overexpressing HEK293T cells in the presence of acetyl-CoA. Acetylation was detected using an anti-pan-acetyl-lysine antibody (upper: experimental setup; lower: Western blot, lane 4). (**F**,**G**) MSL1 enhances MOF-mediated regulation of AURKB. HEK293T cells were co-transfected with Flag-AURKB and MOF, with or without MSL1 overexpression or knockdown. Acetylation, ubiquitination, and H3S10p were assessed by Western blot in both input and Flag IP fractions. (**H**) MSL1 is required for CPC complex integrity. Myc-AURKB and Flag-INCENP were co-expressed in MSL1-KO HEK293T cells. Interaction between AURKB and INCENP, as along with AURKB acetylation, ubiquitination, and downstream H3S10p, was evaluated by Western blot using specific antibodies.

**Figure 4 cells-14-01100-f004:**
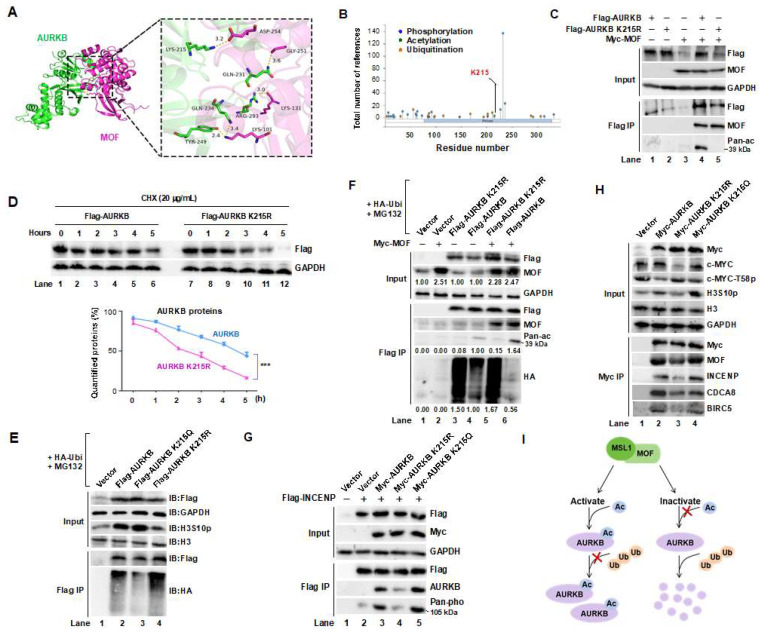
Lysine 215 is the primary acetylation site targeted by the MOF/MSL complex on AURKB in HEK293T cells. (**A**) Three-dimensional docking analysis of MOF (pink) and AURKB (green) identifies their interaction interface, with a magnified view shown in the right panel. Hydrogen bonds between the two proteins are indicated by yellow dashed lines. (**B**) Prediction of AURKB K215 acetylation (red) based on the PhosphoSitePlus^®^ database. (**C**) Co-IP followed by Western blot analysis confirms K215 as the primary acetylation site on AURKB targeted by the MOF/MSL complex. (**D**) Mutation of the K215 significantly reduces AURKB protein half-life. *** *p* < 0.001 compared to AURKB group, Student’s *t*-test. (**E**) Acetylation at K215 is critical for maintaining AURKB stability and kinase activity. (**F**) MOF stabilizes AURKB by acetylating the K215, thereby preventing its degradation. (**G**) The AURKB K215R mutant (acetylation-deficient) retains kinase activity toward its substrate INCENP, comparable to wild-type (WT) and the acetylation-mimetic mutant AURKB K215Q. (**H**) The K215R mutation disrupts AURKB’s ability to form the CPC and reduces the expression of core components, including INCENP, CDCA8, and BIR5. Additionally, the K215R mutant decreases c-MYC levels while increasing phosphorylation of c-MYC at T58, a modification associated with its proteasomal degradation. GAPDH and H3 were used as internal controls. (**I**) The MSL/MOF complex acetylates AURKB at K215, a modification essential for sustaining CPC integrity and AURKB enzymatic activity.

**Figure 5 cells-14-01100-f005:**
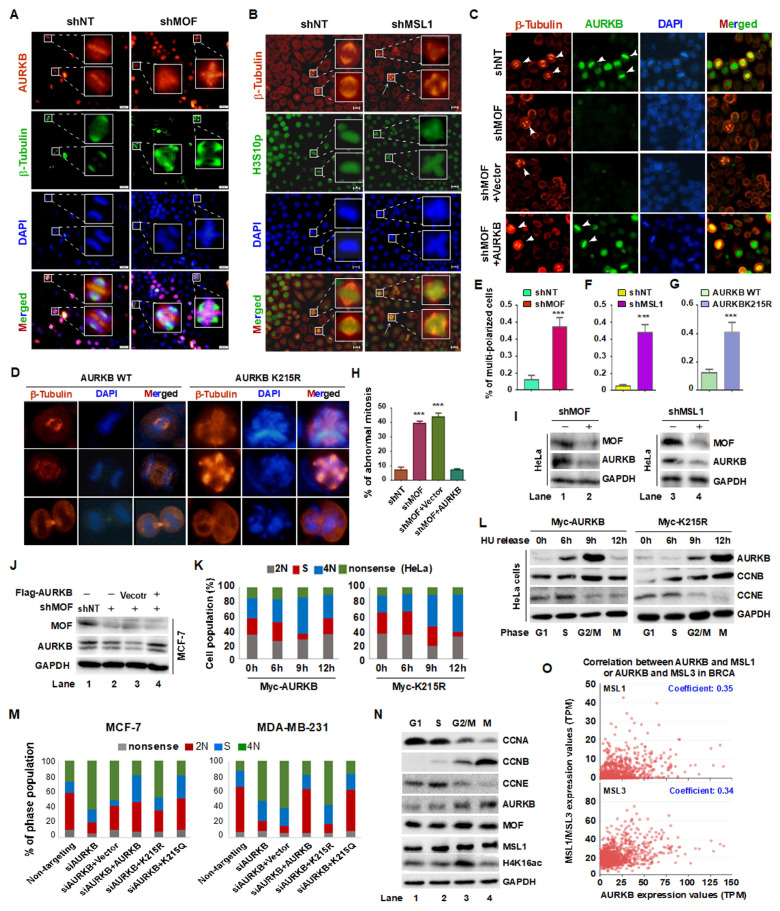
MOF/MSL complex-mediated acetylation of AURKB at K215 regulates G2/M phase progression. (**A**,**B**) IF staining of β-Tubulin (green in A and red in B), AURKB (red), H3S10p (green) in HeLa cells treated with shMOF (**A**) or shMSL1 (**B**). Nuclei were stained with DAPI (blue). Scale bars: 200 μm. (**C**) IF staining of β-Tubulin (red) and AURKB (green) in MCF-7 cells treated with shMOF. Nuclei were stained with DAPI (blue). Scale bars: 100 μm. Arrows represent cells undergoing mitosis. (**D**) IF staining of β-Tubulin (red) in HeLa cells overexpressing wild-type (WT) or K215R-mutant AURKB, nuclei were stained with DAPI (blue). Scale bars: 20 μm. (**E**–**H**) Quantification of spindle multipolar cells corresponding to panels (**A**–**D**), respectively. (**I**,**J**) Validation of MOF and MSL1 knockdown efficiency in the experiments shown in panels (**A**–**C**) via Western blot. (**K**) Transient transfection of the AURKB K215R mutant in HeLa cells delayed cell cycle progression compared to the WT AURKB. Effect of MOF overexpression on AURKB protein levels in MCF-7 and MDA-MB-231 cells. (**L**) Subcellular fractionation of HeLa cells arrested in the S and M phases using hydroxyurea (HU) and nocodazole, respectively. Cytoplasmic and nuclear fractions were separated by centrifugation and analyzed by Western blot. GAPDH was used as a cytoplasmic marker. (**M**) Flow cytometry analysis of cell cycle progression. MCF-7 and MDA-MB-231 cells were transfected with wild-type or mutant AURKB were analyzed in AURKB knockdown backgrounds. (**N**) Co-localization of MOF, MSL1, and AURKB across different cell cycle phases. HeLa cells were synchronized at specific phases: (1) G1 phase by serum starvation (24 h); (2) S phase by HU treatment (1 mM, 24 h); (3) G2/M phase by nocodazole treatment (500 ng/mL, 16 h); (4) M phase by release from nocodazole arrest (1 h). (**O**) Gene expression correlation between AURKB and MSL1 or AURKB and MSL3 in breast cancer. TCGA database, *Y*-axis represents the expression of MSL1 and MSL3, and *X*-axis represents the expression of AURKB. *** *p* < 0.001 compared to shNT or pcDNA3.1 group, Student’s *t*-test.

**Figure 6 cells-14-01100-f006:**
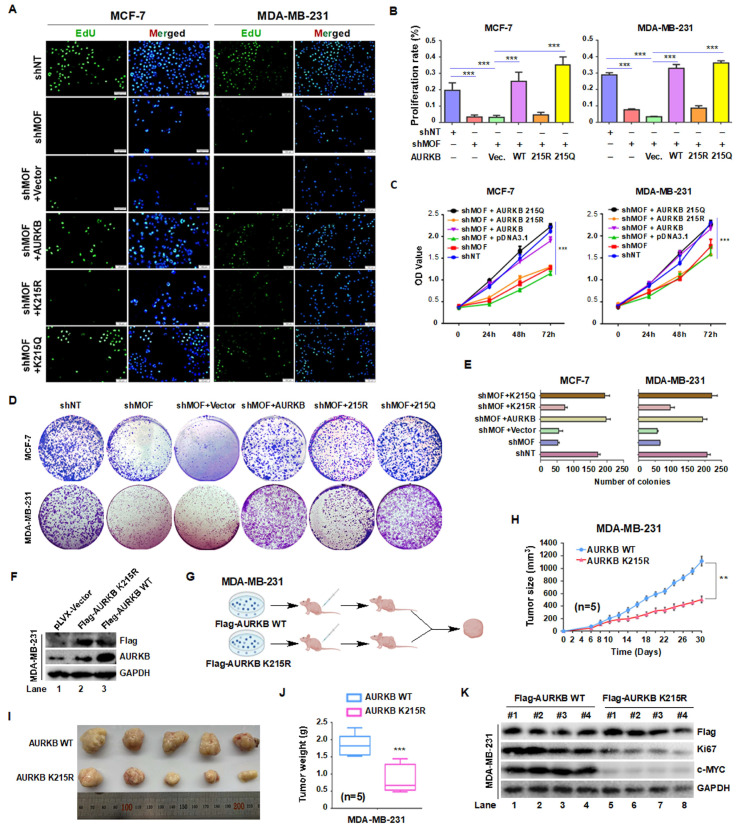
MOF-mediated acetylation of AURKB at K215 regulates the proliferation of MCF-7 and MDA-MB-231 cells. (**A**) EdU incorporation assay to assess cell proliferation. Cells were incubated with EdU for 2 h and analyzed using the BeyoCleck™ EdU Cell Proliferation Kit (C0071s). Scale bar: 50 μm. Green, EdU; blue, stained nuclei with DAPI. (**B**) Quantification of EdU-positive cells. (**C**) CCK-8 assay to evaluate cell viability. ****p* < 0.001. one-way ANOVA. (**D**) Colony formation assays, with colonies stained using crystal violet. (**E**) Quantification of colony formation efficiency. (**F**) Generation of stable MDA-MB-231 cell lines expressing Flag-tagged AURKB or the AKRKB K215R mutant. (**G**) Schematic representation of the animal experiment design. (**H**) Tumor growth progression over time. (**I**) Representative images of tumors at the end of the experiment (*n* = 5 per group). * **p* < 0.01 compared to AURKB group, Student’s *t*-test. (**J**) Tumor weights at the experiments termination in mice injected with AURKB or AURKB K215R-expressing cells. *** *p* < 0.001 compared to the AURKB group. (**K**) Western blot analysis of protein expression in whole-cell lysates from four tumors per group.

**Figure 7 cells-14-01100-f007:**
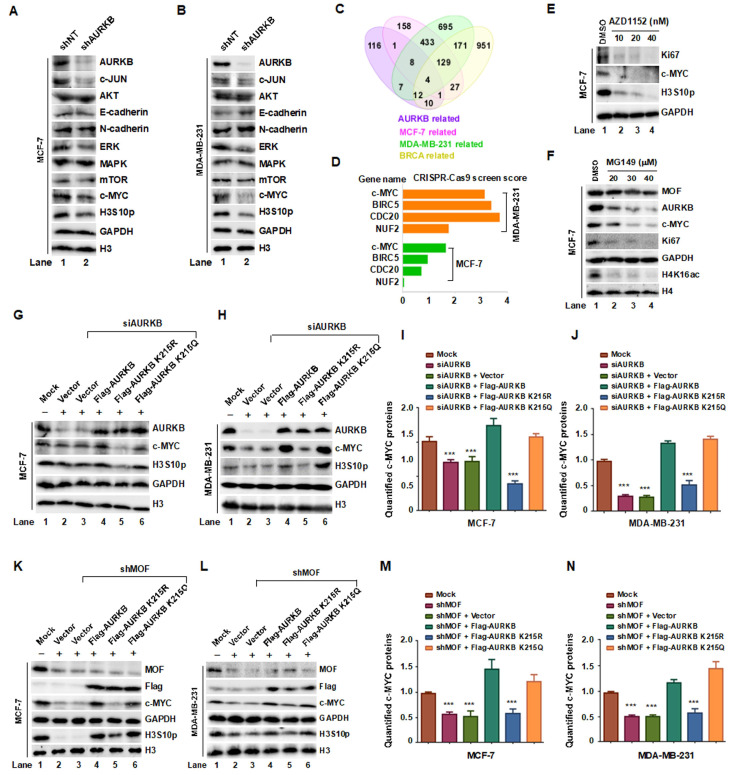
The MOF-AURKB K215ac axis promotes breast cancer cell proliferation by stabilizing c-MYC protein. (**A**,**B**) Western blot analysis of the indicated protein expression levels in MCF-7 and MDA-MB-231 cells transfected with either shNT or shAURKB. (**C**,**D**) Venn diagram showing the intersection of data from BioGRID ORCS, STRING and TCGA Databases, identifying AURKB downstream targets involved in breast cancer proliferation. (**E**) Treatment of MCF-7 cells with the AURKB kinase inhibitor AZD1152 (10, 20, and 40 nM for 24 h) resulted in decreased c-MYC- levels. (**F**) Inhibition of MOF activity using the enzyme inhibitor MG149 reduced the expression of AURKB, c-MYC, and Ki67 in MCF-7 cells. (**G**,**H**) Effects of overexpressing WT, K215R, and K215Q AURKB in MCF-7 and MDA-MB-231 cells on c-MYC levels following AURKB knockdown. GAPDH and histone H3 were used as internal controls. (**I**,**J**) Quantification of c-MYC protein levels. *** *p* < 0.001 compared to non-targeting siRNA group. (**K**,**L**) Effects of overexpressing WT, K215R, and K215Q AURKB on c-MYC, H3S10p, and AURKB levels following MOF knockdown. (**M**,**N**) Quantification of c-MYC protein levels. *** *p* < 0.001 compared to non-targeting shRNA group.

## Data Availability

The data that support the findings of this study are available from the corresponding author upon reasonable request.

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
