# Peer review of "Histone Acetyltransferase MOF-Mediated AURKB K215 Acetylation Drives Breast Cancer Cell Proliferation via c-MYC Stabilization"

_cells, 2025, doi:10.3390/cells14141100_

Round 1
Reviewer 1 Report
Comments and Suggestions for Authors
In their paper, entitled “Histone acetyltransferase MOF-Mediated AURKB K215 Acetylation Drives Breast Cancer Cell Proliferation via c-MYC Stabilization”, the Authors report that the MOF/MSL complex plays a crucial role in mitotic progression by acetylating the AURKB serine/threonine kinases, at the level of K215; this activity stabilizes the chromosomal passenger complex (CPC), by enhancing AURKB kinase activity, in turn ensuring proper spindle assembly and chromosomal alignment during early mitosis.
The paper is of interest and suitable for Cells, also because the Authors performed many kinds of experiments in order to acquire a clear picture of the analyzed events.
I only have two minor comments:
- at line 243, the Authors write that the experiments shown in Fig. 1E-F suggest a post-transcriptional AURKB regulation. Thus, it might be of interest a brief comment on this aspect (that should be worth of further analysis in the future) in the “Discussion” and/or in the “Conclusion” sections, also because, at lines 638-639, they write that “in hepatocellular carcinoma, AURKB mRNA levels are significantly elevated in tumor tissues and serve as an independent prognostic marker for disease aggressiveness”;
- the parts of figures 5 and 6 that show cells are too small in order to see clearly cell conditions.
Author Response
Comments and Suggestions for Authors
In their paper, entitled “Histone acetyltransferase MOF-Mediated AURKB K215 Acetylation Drives Breast Cancer Cell Proliferation via c-MYC Stabilization”, the Authors report that the MOF/MSL complex plays a crucial role in mitotic progression by acetylating the AURKB serine/threonine kinases, at the level of K215; this activity stabilizes the chromosomal passenger complex (CPC), by enhancing AURKB kinase activity, in turn ensuring proper spindle assembly and chromosomal alignment during early mitosis.
The paper is of interest and suitable for Cells, also because the Authors performed many kinds of experiments in order to acquire a clear picture of the analyzed events.
I only have two minor comments:
- At line 243, the Authors write that the experiments shown in Fig. 1E-F suggest a post-transcriptional AURKB regulation. Thus, it might be of interest a brief comment on this aspect (that should be worth of further analysis in the future) in the “Discussion” and/or in the “Conclusion” sections, also because, at lines 638-639, they write that “in hepatocellular carcinoma, AURKB mRNA levels are significantly elevated in tumor tissues and serve as an independent prognostic marker for disease aggressiveness”;
We appreciate the reviewer’s constructive comments. In accordance with the suggestion, we have added the following statement to the Discussion section: “Importantly, our study reveals that the MOF/MSL complex promotes breast cancer cell proliferation by stabilizing the CPC through acetylation of AURKB at K215—without altering AURKB mRNA expression—highlighting the significance of post-translational regulation in tumor progression.”
- The parts of figures 5 and 6 that show cells are too small in order to see clearly cell conditions.
Thank you for the reviewer’s reminder. Appropriate adjustments have been made to the new Figures 5C and 6A. We believe that, compared to the original version, the cell images have been improved and are now clear to a certain extent.

Reviewer 2 Report
Comments and Suggestions for Authors
Miao et al. aimed to present a perspective analysis in their manuscript "Histone Acetyltransferase MOF-Mediated AURKB K215 Acetylation Drives Breast Cancer Cell Proliferation via c-MYC Stabilization," wherein they illustrate that MOF-mediated acetylation of AURKB augments its stability and functionality. MOF-mediated AURKB acetylation enhances c-MYC accumulation, ultimately promoting malignant proliferation in breast cancer cells.
This study details a meticulously conducted investigation showing that the histone acetyltransferase MOF (KAT8) acetylates Aurora kinase B (AURKB) at lysine 215 (K215), therefore stabilizing the kinase by inhibiting ubiquitin-mediated degradation. This alteration enhances the integrity and function of the chromosomal passenger complex (CPC), hence promoting mitotic progression and stabilizing the oncogene c-MYC. The research connects the post-translational control of AURKB to tumorigenic effects in breast cancer models, presenting a significant molecular mechanism with possible clinical implications.
The study is highly original and methodologically rigorous, offering rich functional validation through a comprehensive set of experimental approaches, including CRISPR-mediated gene knockouts, in vitro lysine acetyltransferase assays, cycloheximide chase experiments, and in vivo xenograft models. Notably, the authors have provided well-annotated original images that enhance data transparency and reproducibility, supporting the integrity of their findings. This work is the first to identify lysine 215 (K215) of AURKB as a direct acetylation target of the histone acetyltransferase MOF, establishing a previously unrecognized link between MOF enzymatic activity and AURKB protein stabilization. By elucidating this post-translational mechanism, the study offers a compelling explanation for AURKB overexpression in breast cancer that goes beyond transcriptional regulation, thereby advancing our understanding of tumor cell proliferation and mitotic control.
However, there are several substantial issues that need to be addressed for the manuscript's improvement, listed below.
- Although in vitro HAT inhibition is tested (e.g., MG149), the study does not examine pharmacological inhibition of MOF in the xenograft model. This would strongly enhance translational relevance.
- Some western blots (e.g., Figs. 1G, 2M, 4F) are visually dense or marginally cropped. The inclusion of densitometric quantification (e.g., AURKB/GAPDH ratios) would enhance clarity and facilitate a more objective assessment of protein stability. The WBs can be margined for better visualization.
- While MSL1 is necessary for MOF’s activity, it remains unclear whether it contributes independently to CPC integrity or AURKB regulation. Further clarification or discussion would be beneficial.
- The predicted K215 interaction is supported by docking and database evidence (PhosphoSite), but empirical structural evidence (e.g., mass spectrometry confirmation of K215 acetylation site) would provide definitive validation.
- Improve figure annotations for clarity (e.g., mark lanes corresponding to WT, mutants, and controls in western blots).
- The inclusion of MOF and AURKB expression data from breast cancer patient cohorts (e.g., TCGA) can provide valuable support for this study by contextualizing the findings.
- Discussion on the therapeutic implications of dual targeting MOF and AURKB in resistant breast cancer can be an open platform for future prospects of this study
There is room for grammatical improvement.
Author Response
Reviewer 2:
Comments and Suggestions for Authors
Miao et al. aimed to present a perspective analysis in their manuscript "Histone Acetyltransferase MOF-Mediated AURKB K215 Acetylation Drives Breast Cancer Cell Proliferation via c-MYC Stabilization," wherein they illustrate that MOF-mediated acetylation of AURKB augments its stability and functionality. MOF-mediated AURKB acetylation enhances c-MYC accumulation, ultimately promoting malignant proliferation in breast cancer cells.
This study details a meticulously conducted investigation showing that the histone acetyltransferase MOF (KAT8) acetylates Aurora kinase B (AURKB) at lysine 215 (K215), therefore stabilizing the kinase by inhibiting ubiquitin-mediated degradation. This alteration enhances the integrity and function of the chromosomal passenger complex (CPC), hence promoting mitotic progression and stabilizing the oncogene c-MYC. The research connects the post-translational control of AURKB to tumorigenic effects in breast cancer models, presenting a significant molecular mechanism with possible clinical implications.
The study is highly original and methodologically rigorous, offering rich functional validation through a comprehensive set of experimental approaches, including CRISPR-mediated gene knockouts, in vitro lysine acetyltransferase assays, cycloheximide chase experiments, and in vivo xenograft models. Notably, the authors have provided well-annotated original images that enhance data transparency and reproducibility, supporting the integrity of their findings. This work is the first to identify lysine 215 (K215) of AURKB as a direct acetylation target of the histone acetyltransferase MOF, establishing a previously unrecognized link between MOF enzymatic activity and AURKB protein stabilization. By elucidating this post-translational mechanism, the study offers a compelling explanation for AURKB overexpression in breast cancer that goes beyond transcriptional regulation, thereby advancing our understanding of tumor cell proliferation and mitotic control.
However, there are several substantial issues that need to be addressed for the manuscript's improvement, listed below.
- Although in vitro HAT inhibition is tested (e.g., MG149), the study does not examine pharmacological inhibition of MOF in the xenograft model. This would strongly enhance translational relevance.
We fully understand and appreciate the reviewers’ suggestions. According to current research, MG149 can also act as an inhibitor of the TIP60 acetyltransferase (KAT5). Although MOF-mediated H4K16ac levels were significantly reduced in MG149-treated breast cancer cells, we cannot completely rule out the potential involvement of TIP60 in the mouse xenograft experiments. Therefore, in the mouse tumor transplantation experiments, we focused on investigating the role of MOF-mediated acetylation of AURKB at the K215 site in promoting breast cancer cell proliferation.
- Some western blots (e.g., Figs. 1G, 2M, 4F) are visually dense or marginally cropped. The inclusion of densitometric quantification (e.g., AURKB/GAPDH ratios) would enhance clarity and facilitate a more objective assessment of protein stability. The WBs can be margined for better visualization.
We appreciate the constructive feedback provided by the reviewers. In the revised manuscript, we have performed quantitative analyses of protein expression in Figs. 1G, 2M, and 4F, using GAPDH or Tubulin as internal loading controls. The quantitative results are shown below the corresponding protein bands.
- While MSL1 is necessary for MOF’s activity, it remains unclear whether it contributes independently to CPC integrity or AURKB regulation. Further clarification or discussion would be beneficial.
We agree with the reviewer's comments. Whether MSL1 promotes CPC integrity independently of MOF or is regulated by AURKB cannot be fully determined based on the current experimental results. However, our data support the notion that MSL1, as a scaffold protein, stabilizes the integrity of the MOF/MSL complex and enhances its acetyltransferase activity, thereby facilitating the acetylation of AURKB. Consequently, knockout or downregulation of MSL1 leads to reduced acetylation of AURKB, ultimately decreasing the phosphorylation of its substrate H3S10 by AURKB (Fig 3G&H). The CPC complex consists of AURKB, INCENP, BIRC5, and CDCA8. Acetylated AURKB promotes the phosphorylation of INCENP, thereby enhancing the overall kinase activity of the CPC complex. The relevant content has been added to the Discussion section.
- The predicted K215 interaction is supported by docking and database evidence (PhosphoSite), but empirical structural evidence (e.g., mass spectrometry confirmation of K215 acetylation site) would provide definitive validation.
We fully agree with the reviewer's comments. Conformation of AURKB K215 acetylation through direct protein analysis would indeed provide definitive validation. However, we believe our experimental evidence is sufficient to demonstrate that this site is acetylated by the MOF/MSL complex.
Firstly, in vitro acetyltransferase assays showed that His-tagged AURKB was acetylated by wild-type MOF, but not by the enzymatically inactive MOF G27E mutant.
Second, co-transfection of AURKB and MOF in cells led to an increased pan-acetylation signal of AURKB following immunoprecipitation; however, this acetylation was abolished in the AURKB K215 point mutant.
Third, both molecular docking and evidence from public database (e.g., PhosphoSite) support the likelihood of acetylation at the K215 site.
- Improve figure annotations for clarity (e.g., mark lanes corresponding to WT, mutants, and controls in western blots).
Thank you for the reviewer's reminder. In the revised manuscript, we have improved the figure annotations in the Western blot panels to more clearly distinguish wild type (WT), mutant constructs, and control groups.
- The inclusion of MOF and AURKB expression data from breast cancer patient cohorts (e.g., TCGA) can provide valuable support for this study by contextualizing the findings.
Thank you for the reviewer's suggestion. In response, we analyzed the correlation between AURKB and MSL1, as well as between AURKB and MSL3, using data from the TCGA database. The results have been added to the revised manuscript as the new Figure 5O.
- Discussion on the therapeutic implications of dual targeting MOF and AURKB in resistant breast cancer can be an open platform for future prospects of this study
We fully agree with the reviewer's perspective. Although AURKB inhibition has demonstrated some efficacy in combination therapies for breast cancer, the emergence of drug resistance remains a significant challenge. Therefore, a dual targeting strategy against both MOF and AURKB could offer a promising therapeutic approach to breast cancer by potentially lowering chemotherapy dosages and thereby reducing drug resistance. In this context, we have discussed the combined inhibitory effects on AURKB and emphasized its critical role in drug resistant breast cancer. The relevant content has been added to the Discussion section.
Comments on the Quality of English Language
Thank you for the reviewer's reminder. A native English speaker has carefully reviewed the entire manuscript to correct grammatical and spelling errors.
